

# Investigation for a multi-silique trait in *Brassica napus* by alternative splicing analysis

Liang Chai[*], Jinfang Zhang[*], Haojie Li, Benchuan Zheng, Jun Jiang, Cheng Cui and Liangcai Jiang

Crop Research Institute, Sichuan Academy of Agricultural Sciences, Chengdu, Sichuan Province, China
[*] These authors contributed equally to this work.

## ABSTRACT

**Background**. Flower and fruit development are vital stages of the angiosperm lifecycle. We previously investigated the multi-silique trait in the rapeseed (*Brassica napus*) line zws-ms on a genomic and transcriptomic level, leading to the identification of two genomic regions and several candidate genes associated with this trait. However, some events on the transcriptome level, like alternative splicing, were poorly understood.

**Methods**. Plants from zws-ms and its near-isogenic line (NIL) zws-217 were both grown in Xindu with normal conditions and a colder area Ma'erkang. Buds from the two lines were sampled and RNA was isolated to perform the transcriptomic sequencing. The numbers and types of alternative splicing (AS) events from the two lines were counted and classified. Genes with AS events and expressed differentially between the two lines, as well as genes with AS events which occurred in only one line were emphasized. Their annotations were further studied.

**Results**. From the plants in Xindu District, an average of 205,496 AS events, which could be sorted into five AS types, were identified. zws-ms and zws-217 shared highly similar ratios of each AS type: The alternative 5′ and 3′ splice site types were the most common, while the exon skipping type was observed least often. Eleven differentially expressed AS genes were identified, of which four were upregulated and seven were downregulated in zws-ms. Their annotations implied that five of these genes were directly associated with the multi-silique trait. While samples from colder area Ma'erkang generated generally reduced number of each type of AS events except for Intron Retention; but the number of differentially expressed AS genes increased significantly. Further analysis found that among the 11 differentially expressed AS genes from Xindu, three of them maintained the same expression models, while the other eight genes did not show significant difference between the two lines in expression level. Additionally, the 205 line-specific expressed AS genes were analyzed, of which 187 could be annotated, and two were considered to be important.

**Discussion**. This study provides new insights into the molecular mechanism of the agronomically important multi-silique trait in rapeseed on the transcriptome level and screens out some environment-responding candidate genes.

Corresponding author
Liangcai Jiang, jlcrape@163.com

## INTRODUCTION

Rapeseed (*Brassica napus* L.), an allotetraploid with a complex genome (AACC, $2n = 38$), is the second leading source of vegetable oil globally (*Liu et al., 2015*). The agronomic traits related to rapeseed yield include the pod (silique) number per plant, branch number, and seed weight (*Liu et al., 2015*; *Zhang et al., 2006*; *Li et al., 2015*). We previously reported that zws-ms, a multi-silique rapeseed line (*Chai et al., 2019*), produces three independent pistils and 9 to 10 stamens on the same receptacle in a flower, which consequently leads to the formation of three independent siliques on a carpopodium rather than the single siliques typically observed. Moreover, this trait was found to be affected by the environment, with temperature considered to be the factor most likely to switch on/off the formation of multi-silique.

Temperature is a major environmental factor that regulates various aspects of plant morphology, physiology, and biochemistry, affecting germination, growth, development, and flowering (*Ren et al., 2019*). Fertility in crops such as rapeseed (*Yu et al., 2015*) and rice (*Oryza sativa*) (*Yu et al., 2017*) is affected by temperature. In winter rapeseed lines, although a period of vernalization under low temperature is necessary to initiate flowering, cold stress inhibits growth and development, disturbs metabolism, and causes wilting or even death. Notably, cold stress also induces alternative splicing (AS) in plants (*Palusa, Ali & Reddy, 2007*; *Iida, 2004*).

AS is defined as the mechanism by which primary transcripts are processed into two or more mature isoforms, which enables a single gene to produce diverse protein products (*Pan et al., 2008*; *Sablok et al., 2011*). These proteins differ from each other not only in structure but also possibly in function, subcellular localization, and/or stability (*Huang et al., 2019*; *Chauhan et al., 2019*). AS is common in plants; for example, in *Arabidopsis thaliana*, more than 60% of intron-containing genes undergo AS (*Syed et al., 2012*). Many environmental factors regulate AS events in plants, including $CO_2$ concentration (*Huang et al., 2019*), light (*Godoy et al., 2019*), salt stress (*Ding et al., 2014*), and nutrient deficiencies (*Nishida et al., 2017*). AS not only provides an important source of transcriptomic and proteomic diversity and plasticity for use in natural selection (*Labadorf et al., 2010*), but it also plays specific roles in the response (*Chauhan et al., 2019*) or adaptation to environmental stresses (*Filichkin et al., 2015*). *Guo et al. (2019a)* and *Guo et al. (2019b)* identified four splicing variants of two *BnCYCD3-1-LIKE* genes in *B. napus* and found evidence that their AS may play an important role in the response to environmental stresses. *Xia et al. (2017)* discovered that the AS with intron retention of *EARLY MATURITY8* (*EAM8*) led to early flowering in a barley (*Hordeum vulgare*) landrace; while in shepherd's purse (*Capsella bursa-pastoris*), flowering time varied with changes in the splicing of a *FLOWERING LOCUS C* (*FLC*) homolog (*Slotte et al., 2009*). In addition, the heterologous expression of a vacuolar membrane $Na^+/H^+$ antiporter gene (*SsNHX1*) AS variant from seepweed (*Suaeda salsa*) enhanced the salt tolerance of *Arabidopsis* (*Li et al., 2009*).

As mentioned above, low temperatures switch off the multi-silique trait in zws-ms rapeseed. When zws-ms plants were planted in Xindu, Sichuan Province, China, the multi-silique trait was continuously stable for years; however, when they were grown in

Ma'erkang, Sichuan Province, where the annual average temperature is consistently 7.6 °C lower, the multi-silique trait disappeared and all plants displayed normal siliques (*Chai et al., 2019*). We previously investigated the association of chromosomal regions with this trait, at the genomic and transcriptomic levels, selecting potential candidates from the differentially expressed genes (DEGs) between the multi- and single-silique plants. However, the involvement of post-transcriptional modifications and the mechanisms by which temperature regulates this multi-silique trait remain unclear. AS is often responsive to cold stress in plants (*Iida, 2004*; *Palusa, Ali & Reddy, 2007*) and is a mechanism by which plants perceive temperature fluctuations and modulate the activity of their transcription factors (*Seo, Park & Park, 2013*). In view of the above insights, we analyzed AS using transcriptome sequencing (RNA-seq) in this study. High-throughput RNA-seq technology is a widely used, highly efficient, and is an economical strategy for transcriptomic profiling (*Tong et al., 2013*; *Wang, Gerstein & Snyder, 2009*). It has become increasingly popular because of the following qualities (*Mortazavi et al., 2008*; *Ozsolak & Milos, 2011*); (*Marioni et al., 2008*; *Tong et al., 2013*; *Sultan et al., 2008*): (1) it can be used to detect and quantify the expression of genes, including those expressed at low levels; (2) it can facilitate the annotation of genes and lead to the discovery of novel genes or transcripts; (3) the results are highly reproducible between both technical and biological replicates; and (4) it can detect AS events.

We performed transcriptome sequencing (RNA-seq) on the flower buds of zws-ms and its near-isogenic line (NIL), zws-217, which produces normal single siliques. This facilitated the identification of the AS events in both lines and the analysis of the differentially expressed AS genes and those with line-specific AS events. Combining these data with Gene Ontology (GO) and Kyoto Encyclopedia of Genes and Genomes (KEGG) annotations, we identified likely candidate genes switching on or off the multi-silique trait by altering AS events or transcriptional levels in varied environments.. To the best of our knowledge, this is the first time that the regulation of flower/fruit morphology by AS has been investigated in rapeseed, and our results provide insights into this field more generally.

## MATERIALS & METHODS

### Plant materials and growth conditions

The rapeseed line zws-ms and its NIL, zws-217 (*Chai et al., 2019*), were kept in the Crop Research Institute, Sichuan Academy of Agricultural Sciences, China. Both zws-ms and zws-217 were homozygous for almost all genes, differing from each other only in the multi-silique trait of zws-ms (Fig. 1). The NILs zws-217 and zws-ms were both grown in an experimental field in the Xindu District of Chengdu in the Sichuan Basin, China, under normal environmental conditions. Additionally, the both lines were also grown in Ma'erkang, a mountainous area in western Sichuan, with a much lower annual average temperature. The annual average temperature in Xindu and Ma'erkang is 16.2 °C and 8.6 °C, respectively (*Chai et al., 2019*).

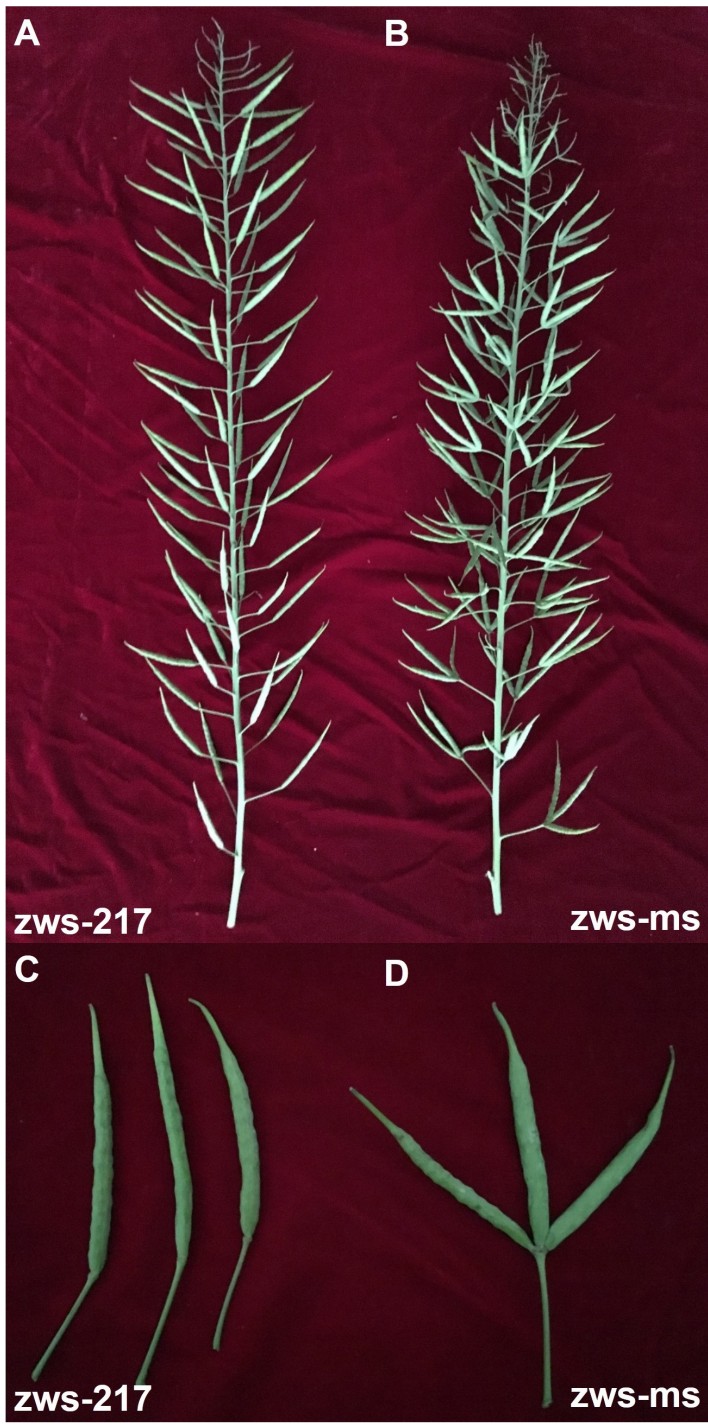

**Figure 1** **Multi-silique trait in zws-ms, compared with the single siliques of its near-isogenic line zws-217.** Multi-silique trait in zws-ms, compared with the single siliques of its near-isogenic line zws-217. (A) Main inflorescence from zws-217; (B) main inflorescence from zws-ms; (C) siliques from zws-217; (D) siliques from zws-ms.

### Total RNA extraction and sequencing library construction

Three zws-ms plants (samples T01, T02, and T03) and three zws-217 plants (T04, T05, and T06) were selected for RNA isolation, as described previously (*Chai et al., 2019*). Flower buds were detached from each plant at the budding stage (BBCH 57), and their total RNA was extracted using an RNA Isolation Kit (Tiangen, Beijing, China). The quality and concentration of the RNA were determined using a NanoDrop 2000 (Thermo Fisher Scientific, Waltham, MA, USA), and the sequencing libraries were generated using an RNA Library Prep Kit for Illumina (New England Biolabs, Ipswich, MA, USA).

### Sequencing and expression analysis

The samples were sequenced on a HiSeq X Ten platform (Illumina, San Diego, CA, USA) and paired-end reads were generated. Low-quality reads and adaptor sequences were removed, and clean reads were used for the following analysis. TopHat2 (*Kim et al., 2013*) was used to map the clean reads onto the *Brassica napus* reference genome (*Chalhoub et al., 2014*) with default parameters "–read-mismatches 2 –read-edit-dist 2 –library-type fr- –max-intron-length 5000000". The number of fragments per kilobase of transcripts per million fragments mapped (FPKM) was calculated to represent the gene expression level, and the DESeq R package (*Anders & Huber, 2010*) was used to analyze the differential expression. The *P*-value was adjusted using Benjamini and Hochberg's approach to control the false discovery rate (FDR). The relative expression levels of each transcript calculated using DESeq were used to define the DEGs, which were defined as having a fold change >4 and an FDR <0.01. Pearson's correlation coefficients were determined for the three biological replicates of each line to determine the reliability of the DEGs. Moreover, real-time quantitative PCR (qPCR) was performed to validate the transcriptome sequencing. Since the validation for transcriptome sequencing data from plants in Xindu had been confirmed previously (*Chai et al., 2019*), we only validated the data from plants in colder Ma'erkang herein. Amplification reactions were performed on iQ$^{TM}$5 Real-Time PCR System (Biorad) as follows: an initial denaturation step at 95 °C for 3 min, 39 cycles at 95 °C for 10 s, 60 °C for 30 s, and 72 °C for 30 s. After each run, a melt curve was acquired to check for amplification specificity by heating the samples from 60 °C to 95 °C. Three biological replicates were applied.

### AS event analysis

The cleaned sequence data were aligned to the reference genome using TopHat2 (*Kim et al., 2013*) with default settings mentioned above. The resultant gapped alignment data in a binary alignment format were then used as an input for Cufflinks and Cuffcompare, which were run using the default settings to assemble the transcripts and identify splicing junctions from the alignment data. For the AS detection and annotation, the AS events were annotated with ASprofile (*Florea, Song & Salzberg, 2013*), which uses Cufflinks and Cuffcompare outputs as input data. Default parameters of the software were used.

### Annotation of genes

Gene function was annotated based on the following databases: Nr (NCBI nonredundant protein sequences), Nt (NCBI nonredundant nucleotide sequences), Pfam (Protein

family), KOG/COG (Clusters of Orthologous Groups of proteins), Swiss-Prot (a manually annotated and reviewed protein sequence database), KO (KEGG Ortholog database), and GO (Gene Ontology).

The GO enrichment analysis of the DEGs was performed using the GOseq R packages based on a Wallenius noncentral hypergeometric distribution (*Young et al., 2010*), which can adjust for gene length bias in the DEGs.

The KEGG database (*Kanehisa et al., 2007*) is a resource used to explore the high-level functions and utilities of the biological system, such as the cell, organism, and ecosystem, from molecular-level information, especially using large-scale molecular datasets generated from genome sequencing and other high-throughput experimental technologies (http://www.genome.jp/kegg/). KOBAS (*Mao et al., 2005*) software was used to test the statistical enrichment of the DEGs in the various KEGG pathways. Default parameters were used.

## RESULTS

### Transcriptome sequencing

Flower buds from three plants of both the multi-silique line zws-ms and the single-silique NIL zws-217 (Fig. 1) were sampled for RNA extraction. The sequencing saturation and cluster analysis of the samples were determined to ensure the validity of the data. In total, 65.6 Gb of clean data were generated, with an average Q30 value of 90.54%. Each sample generated about 36.65 M clean reads with an average GC content of 47.23% (Table S1). The average proportion of total reads mapped to the reference genome for each sample was 73.72% (Table S2). Validation of this transcriptome sequencing data was previously confirmed by qPCR (*Chai et al., 2019*). Similarly, samples from colder area Ma'erkang also generated abundant data, which was validated by comparing the relative transcript levels of eight DEGs in zws-ms and zws-217 by qPCR. The qPCR analysis (Fig. S1) showed that all genes had similar trends in expression as those observed by transcriptome sequencing (described below). Each sample generated about 22.92M clean reads with an average GC content of 46.27%, and Q30 value of 92.95% (Table S3); average proportion of total reads mapped to the reference genome for each sample was 88.87% (Table S4).

### AS Event Identification and Analysis

According to description by *Reddy (2007)*, alternative splicing events were sorted into 5 classes: Alternative 3′ splice site, Alternative 5′ splice site, Exon Skipping, Intron Retention and Mutually Exclusive Exons. The six samples grown in Xindu under normal conditions displayed an average of 205,496 AS events (Table 1; Table S5). The proportions of each AS type were analyzed in both zws-ms and zws-217. The two lines shared highly similar ratios of each AS type, with the alternative 5′ splice site and alternative 3′ splice site types being the most commonly observed, at 43.48% and 42.77% of AS events for both lines, respectively. The mutually exclusive exons type was the next most common (6.61%), followed by the Intron Retention type (5.92%), and the least common types was Exon Skipping, which represented just 1.22% of the AS events (Fig. 2A; Table 1).

**Table 1  Numbers of alternative splicing events in six samples.**

| Area | Line | Sample ID | Alternative 3′ splice site | Alternative 5′ splice site | Exon skipping | Intron retention | Mutually exclusive exons |
|---|---|---|---|---|---|---|---|
| Xindu | zws-ms | T01 | 87699 | 89187 | 2333 | 11981 | 12765 |
| | | T02 | 87603 | 88889 | 2404 | 11201 | 13100 |
| | | T03 | 88055 | 89623 | 2638 | 12184 | 14060 |
| | zws-217 | T04 | 87953 | 89443 | 2567 | 12501 | 13887 |
| | | T05 | 88012 | 89588 | 2581 | 12722 | 13901 |
| | | T06 | 88001 | 89408 | 2549 | 12373 | 13768 |
| Ma'erkang(colder area) | zws-ms | T01 | 48432 | 49501 | 1970 | 14764 | 5299 |
| | | T02 | 46307 | 47031 | 1526 | 9560 | 3696 |
| | | T03 | 46594 | 47558 | 1712 | 12686 | 4464 |
| | zws-217 | T04 | 48204 | 49168 | 1596 | 12722 | 4733 |
| | | T05 | 46902 | 47912 | 1754 | 12194 | 4777 |
| | | T06 | 47759 | 48853 | 1822 | 13718 | 5105 |

**Notes.**

T01, T02, and T03: Buds of three independent zws-ms plants at the budding stage; T04, T05, and T06: Buds of three independent zws-217 plants at the budding stage.
Alternative 3′ splice site: different-size mRNAs are produced depending on the usage of a proximal or distal 3′ splice site; Alternative 5′ splice site: different-size mRNAs are produced depending on the use of a proximal or distal 5′ splice site; Exon Skipping: an exon is either included or excluded from the mRNA; Intron Retention: an intron is either retained or excised in the mRNA, resulting in different-size transcripts; Mutually Exclusive Exons: adjacent exons are spliced in such a way that only one of them is included at a time in the mRNA.

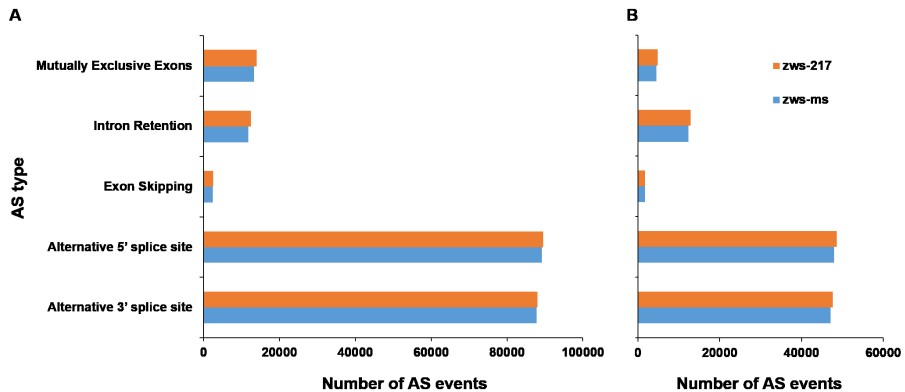

**Figure 2  Statistics of different alternative splicing types of each line.** (A) Data of plants grown in Xindu; (B) data from colder area Ma'erkang. horizontal axis shows the number of each AS type; vertical axis shows different types of AS events.

As to the plants grown in colder area Ma'erkang, the two lines also shared highly similar ratios of each AS type: the alternative 5′splice site and alternative 3′splice site types represented the greatest proportion, at 42.13% and 41.29%, respectively; while the exon skipping accounted for the least proportion 1.51% (Fig. 2B; Table 1). The number of each type of AS events were significantly reduced in colder area Ma'erkang, except for Intron Retention.

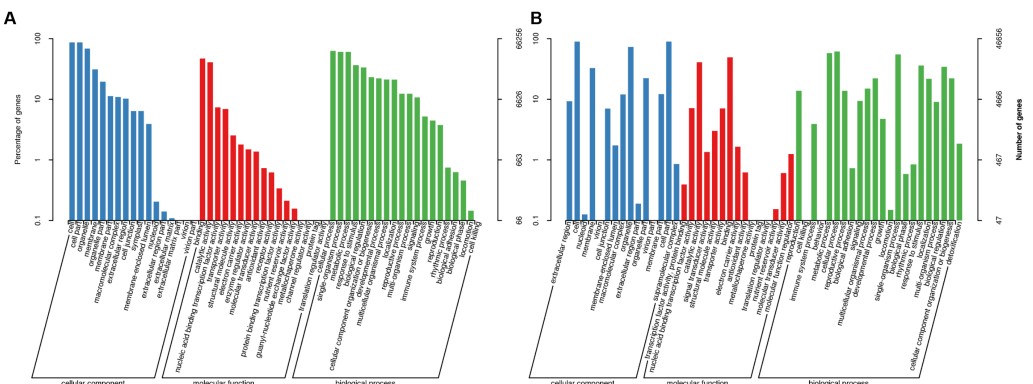

**Figure 3** **Gene ontology (GO) terms associated with the alternatively spliced genes.** GO terms were divided into three categories: biological processes, cellular components, and molecular functions. (A) Data of plants grown in Xindu; (B) data from colder area Ma'erkang. Note: The x-axis shows the GO categories and subclasses of the alternatively spliced genes; the y-axis shows the number or percentage of annotated alternatively spliced genes.

## Annotation of the alternatively spliced genes

To study the biological functions of the genes with AS events, GO and KEGG pathway enrichment analyses were performed. The GO annotations for AS genes from plants in Xindu included 17 terms involved in biological processes (BP; Fig. 3A), 17 terms associated with cellular components (CC), and 20 terms involved in molecular functions (MF). The most highly enriched BP terms observed in the alternatively spliced genes included "cellular process", "single-organism process" and "metabolic process". The most common CC categories were "cell", "cell part" and "organelle". In the MF category, the most enriched terms were "binding", "catalytic activity" and "nucleic acid binding transcription factor activity". Plants grown in Ma'erkang showed highly similar GO data to that in Xindu: 22 terms involved in BP (Fig. 3B), 15 terms associated with CC, and 15 terms involved in MF. Moreover, data from Ma'erkang and Xindu showed the same top-3 most enriched terms in each category.

These KEGG pathways were classified into five major groups: metabolism, genetic information processing, cellular processes, environmental information processing, and organismal systems. Of these, the subgroups "biosynthesis of amino acids", "carbon metabolism", "ribosome", and "RNA transport" contained the highest number of annotated genes (Fig. 4A). Data from Ma'erkang showed similar subgroups containing the most of AS genes: "ribosome", "carbon metabolism", "biosynthesis of amino acids" and "plant hormones signal transduction" contained the highest number of annotated genes (Fig. 4B).

## DEGs with AS and their arabidopsis orthologs

DESeq software was used to identify the different expression levels of the AS genes in zws-ms and zws-217. From Xindu, eleven differentially expressed AS genes were identified, of which four were upregulated and seven were downregulated in zws-ms (Table 2).
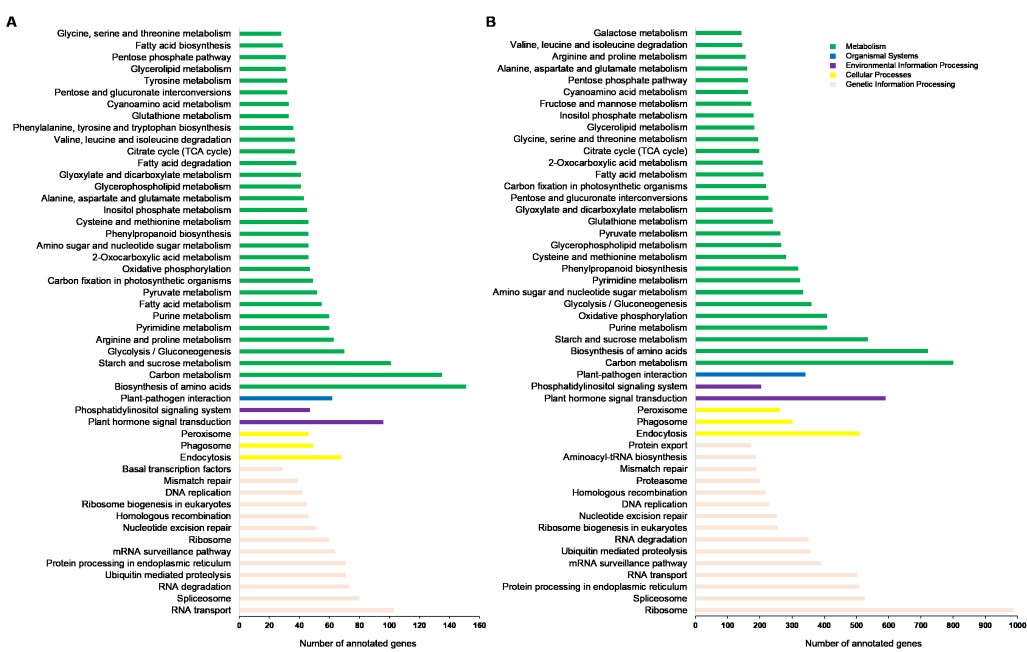

**Figure 4  Classified KEGG pathways associated with the alternatively spliced genes.** The pathways were classified into five major groups: metabolism, genetic information processing, cellular processes, environmental information processing, and organismal systems. (A) Data of plants grown in Xindu; (B) data from colder area Ma'erkang. Note: The x-axis shows the number of annotated alternatively spliced genes; the y-axis shows the pathway categories.

The *Arabidopsis* orthologs of these differentially expressed AS genes were identified using The *Arabidopsis* Information Resource (TAIR; https://www.arabidopsis.org; Table 3). The following orthologs were identified: (1) *AT5G15470*, the ortholog of *BnaA02g02630D*, encodes galacturonosyltransferase 14 (GAUT14); (2) *AT3G15420* encodes the transcription factor TFIIIC (tau55-related protein); (3) *AT1G14800* encodes a nucleic acid-binding, OB-fold-like protein; (4) *AT2G04900* encodes an unknown protein; (5) *AT3G10070* encodes one of two *Arabidopsis* proteins with similarity to the TBP-associated factor, TAF12; (6) *AT1G15060* encodes an alpha/beta hydrolase family protein; (7) *AT3G54620* encodes a bZIP transcription factor-like protein; (8) *AT5G16210* encodes a HEAT repeat-containing protein; (9) *AT3G59000* encodes an F-box/RNI-like superfamily protein; (10) *AT4G16900*, the ortholog of *BnaC07g33980D*, encodes a member of the disease resistance protein (TIR-NBS-LRR class) family; and (11) *AT1G10760* encodes an α-glucan, water dikinase (GWD) required for starch degradation.

When grown in Ma'erkang, the two lines generated increased number of differentially expressed AS genes significantly to 130 (Table S6), including 52 unregulated and 78 down regulated AS genes. Four AS genes were annotated to "response to cold (GO:0009409)": BnaAnng17190D, BnaC01g27600D, BnaC08g39130D and BnaC09g53990D; three AS genes, BnaA07g19340D, BnaC01g27600D, BnaC08g39130D, were annotated to "response to heat (GO:0009408)"; three were related to "response to freezing (GO:0050826)", including BnaA05g28590D, BnaC06g15710D and BnaC08g36010D; BnaCnng24040D was

Chai et al. (2020), *PeerJ*, DOI 10.7717/peerj.10135

**Table 2 Eleven differently expressed alternatively spliced genes and their annotations.**

| Gene ID | Up- or down-regulated | FDR | log$_2$FC | GO annotation | KEGG annotation |
|---|---|---|---|---|---|
| BnaA02g02630D | down | 1.53641E−06 | −2.01793 | Cellular Component: Golgi apparatus (GO:0005794); Biological Process: pollen development (GO:0009555); Biological Process: pollen tube growth (GO:0009860); Molecular Function: polygalacturonate 4-alpha-galacturonosyltransferase activity (GO:0047262); Biological Process: cell wall pectin biosynthetic process (GO:0052325); Cellular Component: pollen tube (GO:0090406); | K13648\|0\|brp:103850984\|probable galacturonosyltransferase 14; K13648 alpha-1,4-galacturonosyltransferase [EC:2.4.1.43] (A) |
| BnaA02g03080D | down | 1.28889E−05 | −2.07335 | – | K15203\|1.23362e−59\|brp:103851045\| uncharacterized LOC103851045; K15203 general transcription factor 3C polypeptide 6 (A) |
| BnaA04g16220D | down | 4.07836E−11 | −2.10257 | – | – |
| BnaA07g04500D | up | 1.84613E−33 | 2.610402 | Cellular Component: mitochondrion (GO:0005739); | – |
| BnaA09g45000D | down | 1.26222E−12 | −2.66637 | Biological Process: RNA splicing, via endonucleolytic cleavage and ligation (GO:0000394); Molecular Function: DNA binding (GO:0003677); Cellular Component: transcription factor TFIID complex (GO:0005669); Biological Process: DNA-templated transcription, initiation (GO:0006352); Biological Process: transcription from RNA polymerase II promoter (GO:0006366); Biological Process: cytokinin-activated signaling pathway (GO:0009736); Biological Process: jasmonic acid mediated signaling pathway (GO:0009867); Biological Process: regulation of ethylene-activated signaling pathway (GO:0010104); Molecular Function: protein heterodimerization activity (GO:0046982); | K03126\|0\|brp:103842750\|transcription initiation factor TFIID subunit 12b-like; K03126 transcription initiation factor TFIID subunit 12 (A) |
| BnaA09g45260D | down | 1.75043E−11 | −2.14465 | Cellular Component: chloroplast (GO:0009507); | – |
| BnaAnng30260D | up | 1.04924E−26 | 4.006055 | Molecular Function: sequence-specific DNA binding transcription factor activity (GO:0003700); Cellular Component: nucleus (GO:0005634); Biological Process: response to xenobiotic stimulus (GO:0009410); Biological Process: response to ethylene (GO:0009723); Biological Process: hormone-mediated signaling pathway (GO:0009755); Biological Process: endoplasmic reticulum unfolded protein response (GO:0030968); Biological Process: positive regulation of transcription, DNA-templated (GO:0045893); Molecular Function: protein heterodimerization activity (GO:0046982); Biological Process: positive regulation of seed maturation (GO:2000693); | – |
| BnaC02g06440D | down | 5.63055E−15 | −2.04283 | Cellular Component: cytoplasm (GO:0005737); | – |
| BnaC06g16950D | up | 4.45014E−05 | 3.19811 | – | – |

Chai et al. (2020), *PeerJ*, DOI 10.7717/peerj.10135

**Table 2** (*continued*)

| Gene ID | Up- or down-regulated | FDR | log$_2$FC | GO annotation | KEGG annotation |
|---------|------------------------|-----|-----------|---------------|-----------------|
| BnaC07g33980D | up | 2.15674E−13 | 2.904839 | Molecular Function: protein binding (GO:0005515); Cellular Component: nucleus (GO:0005634); Cellular Component: mitochondrion (GO:0005739); Cellular Component: cytosol (GO:0005829); Biological Process: response to auxin (GO:0009733); Biological Process: systemic acquired resistance, salicylic acid mediated signaling pathway (GO:0009862); Biological Process: defense response to bacterium (GO:0042742); Molecular Function: ADP binding (GO:0043531); | – |
| BnaC08g49610D | down | 1.65462E−10 | −8.08437 | Molecular Function: protein binding (GO:0005515); Molecular Function: ATP binding (GO:0005524); Cellular Component: mitochondrion (GO:0005739); Biological Process: starch catabolic process (GO:0005983); Biological Process: circadian rhythm (GO:0007623); Cellular Component: chloroplast stroma (GO:0009570); Biological Process: response to symbiotic fungus (GO:0009610); Biological Process: cold acclimation (GO:0009631); Cellular Component: chloroplast envelope (GO:0009941); Biological Process: phosphorylation (GO:0016310); Biological Process: starch biosynthetic process (GO:0019252); Molecular Function: alpha-glucan, water dikinase activity (GO:0050521); | K08244|0|brp:103843262|alpha-glucan water dikinase 1, chloroplastic; K08244 alpha-glucan, water dikinase [EC:2.7.9.4] (A) |

**Table 3** The 11 differently expressed alternatively spliced genes and their orthologs in *Arabidopsis*.

| Gene in *B. napus* | Orthologs in *Arabidopsis* | |
| --- | --- | --- |
| | **Gene ID** | **Description** |
| BnaA02g02630D | AT5G15470 | GALACTURONOSYLTRANSFERASE 14 (GAUT14) |
| BnaA02g03080D | AT3G15420 | Transcription factor TFIIIC, Tau55-related protein |
| BnaA04g16220D | AT1G14800 | Nucleic acid-binding, OB-fold-like protein |
| BnaA07g04500D | AT2G04900 | Unknown protein |
| BnaA09g45000D | AT3G10070 | TBP-ASSOCIATED FACTOR 12 (TAF12) |
| BnaA09g45260D | AT1G15060 | Uncharacterized conserved protein UCP031088, alpha/beta hydrolase |
| BnaAnng30260D | AT3G54620 | BASIC LEUCINE ZIPPER 25 (BZIP25) |
| BnaC02g06440D | AT5G16210 | HEAT repeat-containing protein |
| BnaC06g16950D | AT3G59000 | F-box/RNI-like superfamily protein |
| BnaC07g33980D | AT4G16900 | Disease resistance protein (TIR-NBS-LRR class) family |
| BnaC08g49610D | AT1G10760 | STARCH EXCESS 1 (SEX1), GWD1, SOP1 |

found relevant to "temperature stimulus (GO:0009266)". Moreover, BnaC08g36010D, BnaC08g39130D and BnaC08g39360D were annotated to "regulation of flower development (GO:0009909)", "plant ovule development (GO:0048481)" and "fruit development (GO:0010154)", respectively (Table S6). Compared with the 11 differentially expressed AS genes from normal conditions in Xindu, three of them (BnaA07g04500D, BnaAnng30260D and BnaC06g16950D) maintained the same expression models. In other words, these three genes were upregulated under both normal and colder conditions. While the other 8 genes (BnaA02g02630D, BnaA02g03080D, BnaA04g16220D, BnaA09g45000D, BnaA09g45260D, BnaC02g06440D, BnaC07g33980D and BnaC08g49610D) did not show significant difference between zms-ms and zws-217 in expression level in Ma'erkang.

## Genes with line-specific AS events

Genes with line-specific AS events, defined as those genes with a particular AS event(s) that occurred only in zws-ms or in zws-217, were also identified and analyzed. Unlike the above-mentioned general classifications, we sorted AS events into 12 finer subclasses, in order to identify them more specifically: transcription start site (TSS) and transcription terminal site (TTS) equaled to original Alternative 5′ first exon and Alternative 3′ last exon, respectively; Mutually exclusive exons was subdivided into Alternative exon ends (AE) and Approximate AE (XAE); the original Intron retention was subdivided into single Intron retention (IR), Approximate IR (XIR), Multi-IR (MIR) and Approximate MIR (XMIR); the Cassette exon was then was subdivided into single Skipped exon (SKIP), Approximate SKIP (XSKIP), Multi-exon SKIP (MSKIP) and Approximate MSKIP (XMSKIP). In total, 205 line-specifically expressed AS genes were detected, of which 187 could be annotated (Table S7). Eight genes related to "ovule development", "flower morphogenesis" and other similar processes were highlighted and considered important for further study in the coming future (Table 4, Table S7): (1) An IR event (32749874- 32749905 bp on chromosome C06) of *BnaC06g32640D* occurred line-specifically in zws-217 in Xindu, while this event disappeared in both lines in Ma'erkang. Furthermore, a TSS event (32750089- 32750270
**Table 4** The eight important candidate genes with line-specific alternative splicing events and their orthologs in *Arabidopsis*.

| Gene in *B. napus* | Orthologs in *Arabidopsis* | |
| --- | --- | --- |
| | **Gene ID** | **Description** |
| BnaC06g32640D | AT1G71692 | AGAMOUS-LIKE 12 (AGL12) |
| BnaC07g00780D | AT2G20180 | PHYTOCHROME INTERACTING FACTOR 3-LIKE 5 (PIL5), PHY-INTERACTING FACTOR 1 (PIF1) |
| BnaC04g31460D | AT5G17270 | Protein prenylyltransferase superfamily protein |
| BnaC05g34570D | AT3G18600 | P-loop containing nucleoside triphosphate hydrolases superfamily protein |
| BnaC04g26180D | AT3G54660 | GLUTATHIONE REDUCTASE (GR), EMB2360 |
| BnaC07g25280D | AT3G28730 | HIGH MOBILITY GROUP (HMG), NUCLEOSOME/CHROMATIN ASSEMBLY FACTOR D (NFD), SSRP1 |
| BnaC03g32190D | AT4G00050 | UNFERTILIZED EMBRYO SAC 10 (UNE10), PIF8 |
| BnaCnng68400D | AT5G15020 | SIN3-LIKE 2 (SNL2) |

bp on chromosome C06) only happening in multi-silique zws-ms in Xindu, appeared in both lines when planted in Ma'erkang. This gene was annotated as "vegetative to reproductive phase transition of meristem (GO:0010228)" and "Biological Process: ovule development (GO:0048481)"; (2) *BnaC07g00780D* was associated with "reproductive structure development (GO:0048608)". In Xindu, two specific AE events (1070954-1071329 bp and 1070976- 1071329 bp on chromosome C07, respectively) of it were observed in zws-217, but they both disappeared in two lines when planted in Ma'erkang; (3) *BnaC04g31460D* and (4) *BnaC05g34570D* were related to "regulation of flower development (GO:0009909)". An SKIP event (33329566- 33329615 bp on chromosome C04) for *BnaC04g31460D* and an IR event (33879446-33879532 on chromosome C05) for *BnaC05g34570D* were identified only in zws-217 from Xindu; Similarly, two AE events (27558813–27558932 bp and 27558838–27558932 bp on chromosome C04) for (5) *BnaC04g26180D*, two IR events (31448257–31448346 bp and 31448264–31448353 bp on chromosome C07) for (6) *BnaC07g25280D* and one IR event (19819830–19820358 bp on chromosome C07) for (7) *BnaC03g32190D* were all found line-specificaly in zws-217 from Xindu, and none of them were identified in either liens from Ma'erkang. *BnaC04g26180D* was annotated with "development (GO:0048481)"; *BnaC07g25280D* was annotated as "flower morphogenesis; organ morphogenesis (GO:0009887)" and "vegetative to reproductive phase transition of meristem (GO:0010228)"; *BnaC03g32190D* was annotated as "double fertilization forming a zygote and endosperm (GO:0009567)"; (8) *BnaCnng68400D* was associated with "carpel development (GO:0048440)." An IR and two AE events of this gene were detected specifically in multi-silique line under normal conditions, while in colder area, they were not identified in neither zws-ms nor zws-217.

## DISCUSSION

As an important post-transcriptional metabolic event, AS is involved in many plant growth and developmental processes, such as flowering induction (*Eckardt, 2002*; *Slotte et al.,*

*2009*) and the responses to environmental fluctuations and pathogen attacks (*Barbazuk, Fu & McGinnis, 2008*). To the best of our knowledge, AS events have seldom been reported to regulate the development of flower/fruit morphology in higher plants. This study is the first to analyze the role of AS events in rapeseed flower/fruit development as a whole, let alone those related to the multi-silique trait.

We previously described the morphology and inheritance of the multi-silique trait in *B. napus* (*Jiang et al., 1998*), investigating the associated regions of chromosomes at the genomic level and transcriptomically exploring the DEGs in multi-silique and single-silique plants (*Chai et al., 2019*). The multi-silique trait was found to be controlled by three recessive alleles and was significantly affected by environment; however, the mechanisms by which environmental factors affect this trait remained unknown, even if we knew that temperature could switch on/off the multi-silique trait (*Chai et al., 2019*). As mentioned above, AS is a pathway by which the environment could regulate plant physiology, therefore in this study, we analyzed AS events in order to investigate the mechanism by which plants perceive temperature fluctuations.

In this study, we sampled the buds of three individual plants from zws-ms and zws-217 lines in both Xindu and colder area Ma'erkang, and then subjected them to RNA-seq. All of the four groups generated sufficient data, which was validated by qPCR in earlier and present study successively. The samples in Ma'erkang group generated less data, mainly due to a reduced sequencing depth; nevertheless, this still provided enough data of high quality and assured the accuracy of the subsequent analysis.

We identified all of the genes with AS events in the zws-ms and zws-217 plants. In Xindu group, 11 AS genes were significantly differentially expressed between the multi-silique zws-ms line and its NIL, zws-217, which produces normal siliques. We analyzed their annotations and orthologs in *Arabidopsis*. One such ortholog, *AT5G15470* (also known as Galacturonosyltransferase 14, *GAUT14*), is involved in cell wall pectin biosynthesis (*Caffall et al., 2009*), and the *gaut13 gaut14* double mutant was previously shown to be defective in pollen tube growth (*Wang et al., 2013a*; *Wang et al., 2013b*). *AT3G15420* (the ortholog of *BnaA02g03080D*) and *AT3G10070* (the ortholog of *BnaA09g45000D*) encode subunits of the transcription factor complexes TFIIIC and TAF12, respectively. The former does not appear to be substantially involved in plant development; however, some members of the TAF family are involved in the regulation of morphology. The transgenic expression of *TAF10* from clustered yellowtops (*Flaveria trinervia*) in *Arabidopsis* limited the development of the indeterminate inflorescence and resulted in the production of deformed leaves (*Furumoto et al., 2005*). By contrast, the *taf* mutant in *Arabidopsis* has abnormal phyllotaxis and lacks proper vegetative meristem activity (*Tamada et al., 2007*), indicating the important roles played by the TAFs in plant morphological development. Another DEG AS gene, *BnaA04g16220D*, is not annotated, and its *Arabidopsis* ortholog *AT1G14800* is simply listed as an uncategorized nucleic acid-binding, OB-fold-like protein. The AS gene orthologs *AT2G04900* and *AT1G15060* encode an unknown protein and an uncategorized alpha/beta hydrolase family protein, respectively, so their roles in the regulation of the multi-silique trait are also currently unclear.

Another ortholog for differentially expressed AS gene, *AT3G54620*, is reported to encode a bZIP transcription factor-like protein. Members of this protein family are typically reported to regulate plant tolerance of environmental stresses. The transgenic expression of the maize (*Zea mays*) gene *ZmbZIP72* in *Arabidopsis* enhanced its drought and salt tolerance (*Ying et al., 2012*), while *BnbZIP3*, a ramie (*Boehmeria nivea*) bZIP transcription factor, also increased the drought, salinity, and heavy metal tolerances of transgenic *Arabidopsis* (*Huang et al., 2016*). These genes are also involved in the regulation of other processes; for example, the repression of a bZIP transcription factor gene *OsABI5* expression in rice resulted in low fertility (*Zou et al., 2008*), while the transgenic expression of tomato (*Solanum lycopersicum*) *SlbZIP2* in tobacco (*Nicotiana benthamiana*) increased leaf thickness (*Seong et al., 2016*). To date, however, there are no reports of *bZIP* genes playing a significant role in flower/fruit morphology.

Other AS gene orthologs included *AT5G16210*, encoding a member of the HEAT repeat-containing protein family, which are considered to be involved in intracellular transport (*Hernández-Torres, Jaimes-Becerra & Chomilier, 2014*; *Oeffinger, Dlakic & Tollervey, 2004*). Although *BnaC06g16950D* is not annotated, its ortholog, *AT3G59000*, was identified as encoding an F-box/RNI-like superfamily protein in *Arabidopsis*, which typically function in the plant hormone signaling pathways (*Gao et al., 2009*). Similarly, the ortholog *AT4G16900* encodes a TIR-NBS-LRR class protein, which are known to be involved in disease resistance (*Xun et al., 2019*) and hormonal responses (*Sarazin et al., 2015*). Moreover, *Hewezi et al. (2006)* unexpectedly found that these proteins are associated with developmental abnormalities; transgenic sunflowers (*Helianthus annuus*) expressing the antisense sequence complementing *PLFOR48*, which encodes a TIR-NBS-LRR-type protein, showed stunted growth and a reduction in apical dominance; whereas the pods of transgenic tobacco (*N. tabacum*) lacking *PLFOR48* expression were smaller and showed severe deformations. This indicates that TIR-NBS-LRR-type proteins can regulate the morphology of plants, including fruit morphology, to some extent. Finally, *AT1G10760*, the ortholog of AS gene BnaC08g49610D, which encodes a GWD protein required for starch degradation, is involved in carbohydrate metabolism (*Nadolska-Orczyk et al., 2017*). This gene was also reported to regulate seed size; *Pirone et al. (2017)* found that the length and width of the mature seeds were reduced in the *gwd1 Arabidopsis* mutant, while their density was increased.

To summarize, *AT5G15470*, *AT3G10070*, *AT3G54620*, *AT4G16900*, and *AT1G10760* are all known to be involved in plant development; therefore, their corresponding rapeseed orthologs, *BnaA02g02630D*, *BnaA09g45000D*, *BnaAnng30260D*, *BnaC07g33980D*, and *BnaC08g49610D*, the expression levels of which differed significantly between zws-ms and zws-217, are considered to be potential candidate genes regulating the multi-silique trait.

After that, we continued to investigate data from Ma'erkang, where it is colder and the multi-silique trait in zws-ms line disappeared. Due to the importance of the above-mention 11 differentially expressed AS genes, we first paid attention to them and found that three of them had the same expression models as in Xindu. In other words, they were independent of temperature; in addition, combined with their annotations, they were excluded from the potential candidate genes responding to environmental factors.

On the other hand, other 8 AS genes stopped being differentially expressed between zms-ms and zws-217 in Ma'erkang. That is to say, the expression level of these 8 AS genes (*BnaA02g02630D*, *BnaA02g03080D*, *BnaA04g16220D*, *BnaA09g45000D*, *BnaA09g45260D*, *BnaC02g06440D*, *BnaC07g33980D* and *BnaC08g49610D*) were environment-specific. Besides, *BnaC08g36010D*, *BnaC08g39130D* and *BnaC08g39360D*, which were annotated to flower/ovule/fruit-related terms, these were differentially expressed between zws-ms and zws-217 specifically in Ma'erkang. Moreover, we also found nine temperature-responding AS genes differentially expressed in the colder area: *BnaAnng17190D*, *BnaC01g27600D*, *BnaC08g39130D*, *BnaC09g53990D*, *BnaA07g19340D*, *BnaA05g28590D*, *BnaC06g15710D*, *BnaC08g36010D* and *BnaCnng24040D*. The lower temperature motivated more responding genes and this also explained the reason for increased number of AS genes clustered in each KEGG pathway.

We also explored the line-specific AS genes, which were similarly expressed between zws-ms and zws-217, but contained stable and particular AS event(s) that differed between these two lines. These genes are likely to qualitatively regulate the multi-silique trait. In this case, we could obtain better results by fine-classification of the AS types into 12 subclasses, rather than the five classes mentioned above. Fine classifications could better identify differences between AS types more precisely and subtly. Thus, we found 205 genes of this type, of which 187 could be annotated. Due to the rarity of the multi-silique trait, we did not obtain much useful information from the KEGG pathway analysis. This meant that we were unable to relate this metabolic pathway information to the multi-silique trait directly; however, the GO analysis provided more potential clues. Among these, eight genes were considered to be associated with flower/carpel/ovule development. *BnaC06g32640D* is annotated as being involved in the regulation of the vegetative-to-reproductive phase transition in the meristem (GO:0010228) and in ovule development (GO:0048481). The IR of it in zws-217 seemed to block the multi-silique, while the TSS occurred in zws-ms as if it was positively related to multi-silique trait. Its *Arabidopsis* ortholog, *AT1G71692*, is annotated as *AGAMOUS-LIKE12* (*AGL12*). *Peng et al. (2015)* isolated the *BnFUL* gene in rapeseed, which is homologous to *AGL8* in *Arabidopsis*. Although *BnFUL* was hypothesized to be involved in enhancing pod-shattering resistance, when introduced into *Arabidopsis*, two of the five transgenic plants expressing *BnFUL* unexpectedly had a multi-silique phenotype. However, the mechanisms by which *BnFUL* generates this multi-silique phenotype remain elusive thus far, making the *AGL12* gene identified in this study a potentially important candidate gene. The ortholog gene of *BnaCnng68400D*, *AT5G15020*, encodes an SIN3-LIKE 2 protein (SNL2) known to be important for seed germination or dormancy (*Wang et al., 2016*; *Wang et al., 2013a*; *Wang et al., 2013b*). The zws-ms line-specific AS events of it were detected in Xindu, where the two NILs were distinct from each other in the multi-silique trait, while in Ma'erkang, where the distinguishing trait was switched off, these AS event disappeared in zws-ms. This showed a positive correlation to the trait.

The other six selected line-specific AS genes showed particular AS events in zws-217 in Xindu, implying potential inhibition of the multi-silique morphology. However, these AS events did not occur in zws-ms or zws-217 in Ma'erkang, representing unknown complexity of the mechanisms, which were strongly influenced by environment. Fortunately, their

orthologs in *Arabidopsis* provided many useful clues. *AT2G20180* and *AT4G00050*, orthologs of *BnaC07g00780D* and *BnaC03g32190D* respectively, both encode phytochrome interacting factors (PIFs). Several transcription factors (AP1, SVP, LFY, AG, and SEP3) involved in the regulation of flowering are known to bind to the PIFs, suggesting a direct link with the reported flowering phenotype of the *pif* mutants (*Leivar & Monte, 2014*). *AT5G17270* encodes a prenylyltransferase superfamily protein; however, to the best of our knowledge, there have been no reports about its development-related functions. The *Arabidopsis* ortholog of *BnaC05g34570D* is *AT3G18600*, which encodes a P-loop-containing nucleoside triphosphate hydrolase. While few studies have reported the functions of these proteins, *Liu et al. (2016)* reported that, in sesame (*Sesamum indicum*), one gene encoding a P-loop-containing nucleoside triphosphate hydrolase showed a reduced expression level in sterile buds, indicating that they may play a role in specifying/determining tapetal fate and development. Another line-specific AS ortholog, *AT3G54660*, encodes a glutathione reductase (GR), which was found to increase the fineness (mass per unit length) and bundle strength of cotton (*Gossypium hirsutum*) fiber when transgenically expressed (*Tuttle et al., 2015*). Since cotton fibers are single cells initiating from the epidermis of the outer integument of the ovules (*Ruan et al., 2004*), it can be inferred that GR regulates ovule development to some extent. *AT3G28730*, also known as structure-specific recognition protein *SSRP1*, was also found to regulate floral development, as the *ssrp1-2* mutant *Arabidopsis* produced small and deformed petals with shorter stamens (*Lolas et al., 2010*).

To date, there is some evidence to show that the line-specific AS orthologs *AT2G20180*, *AT4G00050* and *AT3G54660* are related to the regulation of flower/fruit morphology, with clear roles reported for *AT1G71692* and *AT3G28730*. Consequently, their orthologs in rapeseed, *BnaC06g32640D* and *BnaC07g25280D*, respectively, are considered to be important candidate genes regulating the multi-silique trait by conferring or removing some specific line-specific AS events in varied environments. In addition, *BnaCnng68400D*, of which AS events represented a positive correlation to morphology with or without multi-siliques, was also noteworthy.

Some of the genes/loci controlling silique development in *Brassica* plants have previously been reported. In addition to those regulating traits such as the seed weight and silique length (*Liu et al., 2015*) and the number of seeds per silique in *B. napus* (*Li et al., 2015*), some genes related to silique morphology have been cloned and functionally analyzed. *Xiao et al. (2013)* fine-mapped a multi-locular silique gene, *Bjln1*, to a 208-kb region on chromosome A7 in *Brassica juncea* and then revealed that it was the mutations in the CDS and promoter of *BjuA07.CLV1* gene (equivalent to *Bjln1*) to cause the multi-locular trait (*Xiao et al., 2018*). Both *Fan et al. (2014)* and *Yadava et al. (2014)* reported that a mutation in *BrCLV3*, a homologue of *CLAVATA3* in *Arabidopsis*, caused the production of multi-locular siliques in *B. rapa*. However, the multi-silique (or multi-pistil) phenotype of zws-ms is different from the above-motioned multi-locular trait; zws-ms produces three pods on each carpopodium, rather than multiple loculi per pod.

Few studies have investigated this multi-silique trait in rapeseed; however, there have been similar reports of multi-pistil traits in other crops, particularly in wheat (*Triticum aestivum*): *Duan et al. (2015)* discovered a male-sterile wheat mutant, *dms*, with a dwarf

status and multi-pistils, a pleiotropic phenotype found to be controlled by a single recessive gene, which was not identified. *Guo et al. (2019a)* and *Guo et al. (2019b)* reported another multi-ovary trait in the wheat line DUOII, which was controlled by a dominant gene, and used a proteomics approach to propose some candidate proteins. *Yang et al. (2017)* mapped a gene promoting the formation of three pistils (*Pis1*) to chromosome 2D and identified some candidate genes according to their annotations, while *Zhu et al. (2019)* discovered a wheat multi-pistil mutant, *12TP*, which was found to contain a semidominant mutation located on chromosome arm 2DL. Although several studies have explored the multi-pistil trait in wheat, no one has identified any of the specific genes responsible yet.

To sum up, the seven candidate genes mentioned above, including the five differentially expressed AS genes of interest (*BnaA02g02630D*, *BnaA09g45000D*, *BnaAnng30260D*, *BnaC07g33980D*, and *BnaC08g49610D*) and the two genes with line-specific AS events (*BnaC06g32640D* and *BnaC07g25280D*), are therefore hypothesized to regulate the multi-silique trait in rapeseed zws-ms, based on their AS expression levels or line-specific AS events altered by environment. These findings lay a foundation for further functional analyses for future experiments.

## CONCLUSIONS

The utilization of heterosis is a way to increase the yield or improve the quality of crops. Exploring new germplasm resources and genes, as well as clarifying their inheritance, is the foundation of obtaining excellent hybrids. This study provides a novel inspection into the multi-silique trait in rapeseed from the transcriptional perspective by AS responding to environment, deepening the understanding of its molecular mechanism. Further functional verifications are now undergoing study.

### Funding

This research was supported by the National Key Research and Development Plan, grant numbers 2018YFE0108200, 2016YFD0101305 and 2018YFD0100500; the International Cooperation Plan of Sichuan Academy of Agricultural Sciences, grant number CGZH2019GH01; the Modern Agro-Industry Technology Research System of China, grant number CARS-12; the Major Science and Technology Special Subject of Sichuan Province, grant number 2018NZDZX0003; the Scientific Observing and Experimental Station of Oil Crops in the Upper Yangtze River, Ministry of Agriculture, P. R. China, grant number 09203020; the Financial Innovation Ability Promotion Project of Sichuan Province, grant number 2016ZYPZ-013; the Sichuan Science and Technology Program, grant number 18ZDYF0623; and the Sichuan Crop Breeding Community, grant number 2016NYZ0031. The funders had no role in study design, data collection and analysis, decision to publish, or preparation of the manuscript.

### Grant Disclosures

The following grant information was disclosed by the authors:

The National Key Research and Development Plan: 2018YFE0108200, 2016YFD0101305, 2018YFD0100500.

The International Cooperation Plan of Sichuan Academy of Agricultural Sciences: CGZH2019GH01.

The Modern Agro-Industry Technology Research System of China: CARS-12.

The Major Science and Technology Special Subject of Sichuan Province: 2018NZDZX0003.

The Scientific Observing and Experimental Station of Oil Crops in the Upper Yangtze River, Ministry of Agriculture, P. R. China: 09203020.

The Financial Innovation Ability Promotion Project of Sichuan Province: 2016ZYPZ-013.

The Sichuan Science and Technology Program: 18ZDYF0623.

The Sichuan Crop Breeding Community: 2016NYZ0031.

## Competing Interests

The authors declare there are no competing interests.

## Author Contributions

- Liang Chai conceived and designed the experiments, performed the experiments, analyzed the data, authored or reviewed drafts of the paper, and approved the final draft.
- Jinfang Zhang conceived and designed the experiments, performed the experiments, authored or reviewed drafts of the paper, provides the seeds, and approved the final draft.
- Haojie Li, Benchuan Zheng and Jun Jiang and Cheng Cui analyzed the data, prepared figures and/or tables, and approved the final draft.
- Liangcai Jiang conceived and designed the experiments, authored or reviewed drafts of the paper, and approved the final draft.

## Data Availability

Transcriptome Sequencing data are available at NCBI BioProject: PRJNA492226 , SAMN10095249, SAMN10095250, SAMN10095251, SAMN10095253, SAMN10095254, SAMN10095255.

## Supplemental Information

Supplemental information for this article can be found online at http://dx.doi.org/10.7717/peerj.10135#supplemental-information.

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
