# Peer review of "Investigation for a multi-silique trait in Brassica napus by alternative splicing analysis"

_PeerJ, doi:10.7717/peerj.10135_

## Round 0.1 · original submission · Major Revisions

In light of your Appeal, I have looked again at the reviewers`s comments and I am prepared to give you a chance to improve the manuscript. So I am happy to see a thoroughly revised manuscript taking the points of the reviewers into account

Best regards

Dorothea Bartels

· Appeal

Appeal


· · Academic Editor

Reject

Both reviewers found several problems with your manuscript therefore it will be rejected.

·

Basic reporting

no comment

Experimental design

no comment

Validity of the findings

no comment

Additional comments

In this manuscript, Chai et al reported an interesting investigation in genome-wide gene alternative splicing in rapeseed for the muti-silique trait. The authors totally identified about 220168 AS which could be sorted into 12 AS types. Eleven differently expressed AS genes were identified, and their annotations implied that five of these genes including BnFUL were directly associated with the multi-silique trait. Additionally, the 205 stably expressed AS genes were analyzed, of which 187 could be annotated, and three candidates including BnaC06g32640D, BnaC07g25280D, and BnaC01g16410D were considered to be related to the multi-silique trait. These findings may be very helpful for our understanding of muti-silique formation in rapeseed.

Some comments:

In the manuscript, authors mentioned that AS is a pathway by which the environment could regulate plant physiology, therefore in this study, we analyzed AS events in order to investigate the mechanism by which plants perceive temperature fluctuations. However, I could not see it from their data because they did RNA-seq using NILs zws-217 and zws-ms growing in the normal environmental conditions . Low temperatures can switch off the multi-silique trait in zws-ms rapeseed, I suggest authors should also do the RNA-seq at different temperature condition using zws-ms rapeseed.

Reviewer 2 ·

Basic reporting

This manuscript is to investigate how alternative splicing may contribute to a multi-silique trait in Brassica napus by comparing the transcriptome difference between two NIL lines: one has such a trait, while the other does not. The authors previous found the genomic region and candidate genes associated with this trait. In this manuscript, the authors performed an RNAseq experiment and reported their data analysis in regards to all the transcripts that exhibited alternative splicing in this experiment, differentially expressed genes between the two NIL lines that also exhibited alternative splicing in this experiments, and line-specific alternatively spliced genes which the author defined as “genes with stable AS events”. The introduction provided an adequate overview of prior work that support the rationale of this study.

Experimental design

The RNAseq experiment was performed with three biological replicates of each comparison group, which is standard. However, there is a substantial lack of information and experimental confirmation as listed below:
1. Currently the manuscript does not include sufficient AS analysis results to support the conclusion. Supplementary tables need better title and column headers to provide more detailed information. For example, in Table S3, column G has a header of “event pattern”, while the actual cells only listed the coordinates of each splicing event. This does not provide any information as to the read count of each AS event. Information from all six samples should be analyzed and aggregated into one table to show a comparison across samples that directly support descriptions of AS distributions in Result section. The fact that the authors only described any genes that exhibit AS in any samples without their abundance and analysis on how much changes there may be between the two lines does not provide much insight in how AS may contribute to the multi-silique trait. Authors did mention eleven differentially expressed AS genes. However, this diminishes the value of alternative splicing study because changes in ratios of differently spliced isoforms of the same gene may be important in regulating protein abundance and function without changing total transcript level of that gene.
2. In addition, the authors reported 205 “stably expressed AS genes” which was later explained as “with a particular AS events that occurred only in zws-ms or in zws-217”. The term “Stably expressed AS genes” is not self-explanatory. This group of events are basically line-specific AS events according to this description. They could be interesting to have a deeper look and validated by rtPCR. Unfortunately, the authors did not validate any of them. Table S9 only provided their annotations without their abundance or any statistical measures.
3. The description in “AS Event Analysis” and “Annotation of Genes” lacks details regarding statistical parameters, statistical tests used and FDR used, etc.
4. There was no qPCR validation of differentially expressed AS genes. In table 2, the eleven differentially expressed AS genes are only reported as upregulated or downregulated without any fold change and statistics in there. These should be added.
5. Classes of alternative splicing events have been long established in the past (Reddy. Annu. Rev. Plant Biol. 2007. 58:267-94). The authors should use the existing classification and integrate theirs as subclasses.

Validity of the findings

Without enough quantitative and validated results and appropriate analyses on these AS events, the conclusions of this manuscript are weak.

---

## Round 0.2 · Major Revisions

The main problem is that the RNA Seq data were not properly evaluated. This ineeds to be done. There are also some additional points which need revision, please see comments of the reviewer.

Reviewer 2 ·

Basic reporting

The newly added RNA-seq data from a colder condition was not analyzed and discussed.

Experimental design

no comments

Validity of the findings

The authors did not validate RNA-seq results.

Additional comments

1. It is good that the authors included a different set of data from a colder condition. But this set of data needs to analyzed and discussed in this manuscript. It can’t be just there without any contribution to the manuscript.
2. The authors completely ignored the question on validation of RNA-seq data. Table 2 provides logFC from RNA-seq. That is reporting the same information, not validation.
3. Lines 119-122: Please specify annual average temperatures in both Xindu and Ma’erkang.
4. Line 155. Although there is a statement that default parameters were used. It is very cryptic and does not offer any specific information. Please specific statistical tests used and FDR used as asked earlier.
5. There is no Table 5, while it was referred to on line 190. Is it supposed to be Table S5?
6. Line 220: It is differentially expressed, not differently expressed. This mistake appeared again later in the manuscript (Line 315).
7. Lines 263-273: Since new classifications of AS events are created here, there need to be some sort of illustration to visualize the difference between the establish classification and this new one. Since Table S7 is mentioned here. What is the FDR for in Table S7? Column headers need explanations for what they represent in all tables.
8. Lines 275-276: There is a statement saying that 10 genes were selected for further study. But there is no information on what this further study is and what the results of the further study are.

---

## Round 0.3 · Minor Revisions

Please check the language of the manuscript very carefully and correct grammatical and linguistic errors.

Reviewer 2 ·

Basic reporting

The authors have made significant improvement on the manuscript. I am fine with the manuscript with two minor revisions:

1. the manuscript still has grammar errors, such as line 79 "did not showed". The authors should do read the entire manuscript thoroughly and correct all language errors.

2. For the added qPCR results in Figure S1, there is no mentioning of basic experimental conditions, such as the use of biological replicates and how many? the description of methods in manuscript should have sufficient information to allow readers to understand the design of the experiments. It should not be assumed that every paper uses the same conditions.

Experimental design

ok.

Validity of the findings

ok.

---

## Round 0.4 · Minor Revisions

Dear authors,

Your manuscript still needs some further revision before it will become acceptable for publication. Please consider the review below from the Section Editor, Gerard Lazo, and revise your manuscript taking the comments into account:

Best regards

Dorothea Bartels

Section Editor comments:

> The manuscript appears somewhat incomplete as I can see that there are apparent post-transcriptional processing of candidate genes in two distinct environments, but there does not appear to be any contextual sequence examples demonstrating the alternate splicing observed. I would have expected some examples of splice variants of the candidates represented in their contextual setting. As there was no follow-up in an experimental role a preview of the candidate model would go far to bring sense to the findings. Pointing to the candidates is good in one sense, and hypothesizing a role is another, but the evidence provided here still appears too conceptualized. With the culmination of numerous candidates, is there speculation of the more probable candidates and a model representation that can be followed up upon? Pointing to the reference gene ID without representation of the alternate splicing observations is not a helpful lead to the reader. I would propose some additional splicing observations in the context of the environments be supplied. I can see where the manuscript is going, but there is a need for additional information.

---

## Round 0.5 · accepted · Accept

The authors have added additional information and therefore the article becomes acceptable.